# Melanoma Glycome Regulates the Pro-Oncogenic Properties of Extracellular Galectin-3

**DOI:** 10.3390/ijms26104882

**Published:** 2025-05-19

**Authors:** Norhan B. B. Mohammed, Rajib K. Shil, Charles J. Dimitroff

**Affiliations:** 1The Ronald O. Perelman Department of Dermatology, NYU Grossman School of Medicine, New York, NY 10016, USA; Norhan.Mohammed@nyulangone.org; 2Department of Cellular and Molecular Medicine, Herbert Wertheim College of Medicine, Florida International University, Miami, FL 33199, USA; rshil003@fiu.edu

**Keywords:** Galectin-3, GCNT2, glycans, melanoma, metastasis

## Abstract

Metastatic melanoma is an aggressive skin cancer with a five-year survival rate of only 35%. Despite recent advances in immunotherapy, there is still an urgent need for the development of innovative therapeutic approaches to improve clinical outcomes of patients with metastatic melanoma. Prior research from our laboratory revealed that loss of the I-branching enzyme β1,6 N-acetylglucosaminyltransferase 2 (GCNT2), with consequent substitution of melanoma surface I-branched poly-N-acetyllactosamines (poly-LacNAcs) with i-linear poly-LacNAcs, is implicated in driving melanoma metastasis. In the current study, we explored the role of galectin-3 (Gal-3), a lectin that avidly binds surface poly-LacNAcs, in dictating melanoma aggressive behavior. Our results show that Gal-3 favors binding to i-linear poly-LacNAcs, while enforced GCNT2/I-branching disrupts this interaction, thereby suppressing Gal-3-dependent malignant characteristics, including extracellular signal-regulated kinase/mitogen-activated protein kinase (ERK/MAPK) pathway activation, *BCL2* expression, cell proliferation, and migration. This report establishes the crucial role of extracellular Gal-3 interactions with i-linear glycans in promoting melanoma cell aggressiveness, placing GCNT2 as a tumor suppressor protein and suggesting both extracellular Gal-3 and i-linear glycans as potential therapeutic targets for metastatic melanoma.

## 1. Introduction

Melanoma metastasis to distant organs represents a critical challenge in dermato-oncology, accounting for approximately 80% of skin cancer-related fatalities [1]. Although new targeted and immunotherapeutic agents have revolutionized the treatment landscape for metastatic melanoma since 2011, challenges remain in predicting patient response, managing side effects, and overcoming drug resistance [2,3]. Continued research into melanoma cell biology and its interactions with the surrounding tumor microenvironment (TME) will greatly advance our understanding of the mechanisms driving melanoma progression and support the development of new therapies [4].

Cumulative evidence demonstrates that the altered glycosylation profile of cancer cell surface glycoconjugates contributes to the acquisition of cancer hallmarks, which are essential for malignant transformation and progression [5,6]. Prior studies on melanoma illustrate that aberrant sialylation and fucosylation patterns of surface glycoproteins and the predominance of the large tri- and tetra-antennary N-linked glycans (N-glycans) intervene in the progression from normal melanocyte to metastatic phenotype [7]. Moreover, blood group I-antigens (branched poly-N-acetyllactosamines (poly-LacNAcs)) synthesized by I-branching β1,6 N-acetylglucosaminyltransferase 2 (GCNT2) are reported to be preferentially expressed on normal melanocytes, whereas metastatic melanoma cells display mainly i-antigens (linear poly-LacNAcs) with significant downregulation of the I-branching enzyme GCNT2 compared to normal melanocytes [8,9].

Galectins are carbohydrate-binding proteins that are categorized into three groups according to their quaternary structures: prototypical, chimera-type, and tandem-repeat galectins [10]. Galectins are widely expressed by a diverse array of human cell types and are found in the nucleus, cytoplasm, and extracellular space [11]. They bind specifically to β-galactoside-containing glycans, including poly-LacNAc antennae generated on cell surface or extracellular glycoconjugates through the action of variable glycosyltransferase enzymes [10]. The role of galectins in melanoma initiation and progression has been extensively studied [12,13]. However, the specific interaction of galectins with either the GCNT2/I-branched or i-linear glycans and their influence on melanoma cell behavior have not been thoroughly investigated [14]. Galectin-3 (Gal-3) is a chimeric galectin composed of a single carbohydrate recognition domain (CRD) linked to a collagen-like N-terminal domain [15]. This distinctive structure allows the extracellular oligomerization of Gal-3 monomers into oligomers through their N-terminal domains, forming lattice-like structures by bridging between adjacent cells or between cells and extracellular matrix (ECM) [16]. The binding of Gal-3 to cell surface glycosylated proteins such as integrins and epidermal growth factor receptor (EGFR) and ECM proteins like laminin and fibronectin has been reported to play a crucial role in regulating various biological processes [17,18,19].

In this study, we explored the influence of the GCNT2/I-branched glycans on melanoma cell behavior in light of their interaction with extracellular Gal-3. Our goal was to elucidate how GCNT2 expression and the resulting glycosylation patterns impact Gal-3 binding and the subsequent signaling pathways that contribute to melanoma progression. We found that GCNT2 expression in both human and murine melanomas is inversely associated with metastatic potential. Using melanoma cells with enforced GCNT2 expression, data demonstrated that high GCNT2 levels impede Gal-3 binding to its surface receptors, with the glycoprotein receptor β1 integrin serving as a representative ligand model. This reduced binding attenuated the activity of downstream extracellular signal-regulated kinase–mitogen-activated protein kinase (ERK-MAPK) signaling, pro-survival gene expression, cell proliferation, and migration abilities compared to control cells. Collectively, our findings provide critical insights into the glycosylation-dependent mechanisms regulating melanoma progression and highlight the protective role of GCNT2/I-branching in mitigating Gal-3-mediated aggressive disease.

## 2. Results

### 2.1. GCNT2 Is Downregulated in Human and Murine Metastatic Melanomas Compared to Primary Melanomas

Analysis of The Cancer Genome Atlas (TCGA) data revealed that GCNT2 levels are significantly higher in primary melanoma samples (*n* = 103) than in metastatic cases (*n* = 368) (*p* < 0.05) (Figure 1a). Similarly, the interrogation of the RNA sequencing (RNA-seq) dataset GSE122789 showed decreased GCNT2 expression in the highly metastatic B16/BL6 murine melanoma cell line relative to its parental, less metastatic B16 cell line (Figure 1b). These findings reinforce our previously reported observation of the inverse relationship between GCNT2 expression and the metastatic capacity of melanoma [8].

### 2.2. I-Branching Hinders Gal-3 Binding to Its Ligands on the Melanoma Cell Surface

Since the glycan profile of a cancer cell is a major determinant of Gal-3 binding activity [20], and because GCNT2 expression was shown to influence melanoma cell behavior by controlling its growth, signaling, and survival activities [8], we investigated the possibility that GCNT2/I-branching functions as a putative inhibitor of Gal-3 binding activity. Flow cytometry analysis revealed a significant reduction in Gal-3 binding to A375 GCNT2-overexpressing (GCNT2^OE^) cells compared to empty vector control (EVCtrl) cells (*p* < 0.05) (Figure 1c), suggesting a predilection of Gal-3 to bind i-linear over I-branched poly-LacNAcs. Furthermore, Gal-3 binding was significantly reduced following a 48 h treatment with the N-glycosylation inhibitor Kifunensine to approximately 26% and 50% of untreated binding levels in EVCtrl and GCNT2^OE^ cells, respectively (*p* < 0.001 and *p* < 0.001, respectively) (Figure 1d). This reduction highlights the critical role of N-glycosylation in mediating Gal-3-glycan interactions on the melanoma cell surface. To further validate these findings, we utilized Gal-3 affinity chromatography followed by immunoblotting for β1 integrin, a well-characterized surface receptor for Gal-3 on melanoma cells [21,22]. The results indicate a significantly higher ability of Gal-3 to pull down β1 integrin expressed on EVCtrl cells compared to GCNT2^OE^ cells (*p* < 0.01) (Figure 1e). These data indicate that Gal-3 binding to its glycoprotein receptor β1 integrin on melanoma cells is regulated by the branching activity of the GCNT2 enzyme.

### 2.3. I-Branching Attenuates Gal-3-Dependent Malignant Characteristics of Melanoma Cells

The Mitogen-Activated Protein Kinase (MAPK) pathway, frequently activated in melanoma, functions downstream of β1 integrin-mediated signaling and plays a pivotal role in regulating tumor cell proliferation, migration, and invasion [23,24,25,26]. Given its importance in melanoma progression, we explored whether Gal-3 across various concentrations (0.02, 0.1, 0.5, and 1 μg/mL) influences this pathway and how its effects differ in the presence or absence of GCNT2/I-branching. Data analysis using two-way ANOVA revealed a significant interaction between cell line and Gal-3 treatment on ERK phosphorylation (pERK) (*p* < 0.05). Post hoc tests showed that in A375 EVCtrl cells, Gal-3 significantly increased ERK phosphorylation at concentrations of 0.5 μg/mL (*p* < 0.05) and 1 μg/mL (*p* < 0.001) (Figure 2a). Conversely, in A375 GCNT2^OE^ cells, Gal-3 treatment did not significantly alter ERK phosphorylation at any tested concentration (*p* > 0.05) (Figure 2a). These results indicate that Gal-3 more effectively stimulates ERK phosphorylation in A375 EVCtrl cells compared to A375 GCNT2^OE^ cells, suggesting that the presence of GCNT2/I-branching modulates the intracellular response of the MAPK signaling axis to the exogenous binding of Gal-3 to melanoma cells.

Furthermore, Gal-3-treated EVCtrl exhibited significantly higher expression levels of the pro-survival gene *BCL2* compared to untreated cells (*p* < 0.01), whereas no significant difference was observed in *BCL2* expression between Gal-3-treated and untreated GCNT2^OE^ cells (Figure 2b). The expression levels of the pro-survival gene *BCL2L1* and the proapoptotic genes *BAD* and *BAX* remained unchanged following Gal-3 treatment in both EVCtrl and GCNT2^OE^ cells (Figure 2b). To further investigate the functional effects of Gal-3 treatment on melanoma cells and how they are modulated by I-branching, we evaluated the proliferative abilities of Gal-3-treated cells. Gal-3 treatment significantly increased cell proliferation over time (*p* < 0.001), with a significant interaction between cell line, treatment, and time (*p* < 0.001), indicating that the proliferative response to Gal-3 differed between groups. At 72 h, Gal-3-treated A375 EVCtrl cells showed a significantly higher proliferation compared to untreated A375 EVCtrl cells (*p* < 0.05) (Figure 2c). In contrast, Gal-3 treatment did not significantly enhance proliferation in A375 GCNT2^OE^ cells at any time point (Figure 2c). These findings suggest that Gal-3 preferentially enhances the proliferation of melanoma cells exhibiting reduced levels of surface GCNT2/I-branched glycans. We also evaluated the migratory abilities of Gal-3-treated cells. Our results reveal that Gal-3 significantly enhanced the migration capacities of both EVCtrl and GCNT2^OE^ cells (*p* < 0.01 and *p* < 0.01, respectively) compared to their respective untreated cells (Figure 2d). Importantly, Gal-3-treated EVCtrl cells exhibited greater migratory ability than Gal-3-treated GCNT2^OE^ cells (*p* < 0.05) (Figure 2d), suggesting that I-branching glycosylation attenuates the pro-migratory effects of extracellular Gal-3 on melanoma cells.

## 3. Discussion

This study provides compelling evidence for the role of GCNT2-mediated I-branching glycosylation in modulating Gal-3 interactions and subsequent cellular behaviors in melanoma. Our data demonstrate that GCNT2 expression is inversely correlated with the metastatic potential of human and murine melanomas, reinforcing our previously reported findings of GCNT2 as a suppressor of melanoma progression [8,27]. Our understanding has further evolved to recognize that low GCNT2 levels promote melanoma aggressiveness by enhancing galectin binding (Figure 3). Here, we report that GCNT2/I-branching significantly impairs Gal-3 binding to melanoma cell receptors, particularly β1 integrin, and diminishes Gal-3-induced ERK phosphorylation, *BCL2* expression, cell proliferation, and migration. These results collectively highlight the intricate interplay between glycosylation patterns and Gal-3 signaling in melanoma progression, opening new avenues for potential therapeutic interventions targeting this axis.

While increased expression of β1,6GlcNAc-branched N-glycans, synthesized by β1,6-acetylglucosaminyltransferase V (MGAT5, also known as GnT-V), has been consistently linked to enhanced galectin binding and tumor progression [7,10], our study reveals a contrasting effect of β1,6 branching at the level of poly-LacNAc chains. The differential impact of branching at distinct levels of glycan structures on galectin binding and the resulting cellular behaviors is probably attributed to the altered abundance and accessibility of galectin ligands within the glycocalyx [28,29,30]. MGAT5-mediated branching promotes the formation of tri- and tetra-antennary complex N-glycans [29,31], which are further elongated with poly-LacNAc chains, serving as high-affinity ligands for galectins, including Gal-3 [28]. In contrast, GCNT2-mediated I-branching introduces β1,6-linked N-acetylglucosamine residues to the internal galactose of linear poly-LacNAc chains [9,32]. This modification likely disrupts the spatial presentation and accessibility of galactose residues required for optimal Gal-3 binding, thereby impairing Gal-3 ability to engage glycoprotein receptors and suppressing downstream pro-tumorigenic signaling pathways, including ERK activation and BCL-2-mediated survival mechanisms, as observed in our study. Similarly, the modulation of melanoma cell proliferation and migration by GCNT2/I-branching can be attributed, at least in part, to the altered interactions between Gal-3 and its surface receptors, particularly integrins. Integrins are extensively glycosylated transmembrane proteins with numerous N-glycosylation sites [33], making them major ligands for Gal-3 [21]. Gal-3-mediated cross-linking of integrins has been shown to induce lamellipodia formation and promote migration in melanoma cells [21,34]. Moreover, changes in integrin glycosylation patterns have been shown to significantly impact melanoma cell behavior and metastatic potential [35].

Our observations align with recent studies showing that the tandem-repeat galectin Gal-8 preferentially binds to melanoma cells with reduced GCNT2/I-branching and elevated levels of i-linear poly-LacNAcs, resulting in enhanced activity of pro-tumorigenic signaling pathways [14]. Together with our data, these insights further support the notion that GCNT2/I-branching serves as a suppressive mechanism against galectin-driven malignancy. Of note, the distinct roles of intracellular and extracellular galectins add complexity to their function in melanoma progression [12,36]. While our study shows that GCNT2/I-branching impairs extracellular Gal-3 binding and signaling, our previous work suggests that intracellular Gal-3 exerts opposing effects by modulating transcriptional programs to suppress melanoma metastasis, such as downregulating nuclear factor of activated T-cells 1 (NFAT1) [37]. This dichotomy highlights the need for further research into how galectin localization and glycosylation collectively influence melanoma progression. Future research should also focus on elucidating the upstream regulatory mechanisms governing GCNT2 expression, exploring its potential as a biomarker for disease staging or treatment response, and investigating the broader interplay between GCNT2-mediated glycosylation and other galectins beyond Gal-3 and Gal-8 to better understand their collective impact on melanoma progression and metastatic behavior.

## 4. Materials and Methods

### 4.1. Gene Expression Data Collection and Processing

Transcriptome data for skin cutaneous melanoma (SKCM) were sourced from The Cancer Genome Atlas (TCGA), analyzed, and visualized using the UCSC Xena platform (http://xena.ucsc.edu, accessed on 15 May 2025) [38] as previously described [37]. After excluding samples lacking type or GCNT2 expression, as well as those classified as solid tissue normal or additional metastatic, 471 cases (103 primary, 368 metastatic) were included in the study. Data were generated on the Illumina HiSeq 2000 platform and reported as log2 (value  +  1) RSEM-normalized counts. The RNA-seq dataset GSE122789 (which included 2 murine melanoma cell lines: the parent B16 cell line and the highly metastatic B16/BL6 subline) was analyzed using GREIN (http://www.ilincs.org/apps/grein/?gse=, accessed on 15 May 2025) [39] as previously described [37]. Significance was set at an adjusted *p*-value < 0.05.

### 4.2. Cell Lines and Cell Culture

A375 human melanoma cells were obtained from ATCC, and GCNT2-overexpressing (GCNT2^OE^) and empty vector control (EVCtrl) lines were generated via lentiviral transduction as previously reported [8]. Cells were cultured in Dulbecco’s Modified Eagle Medium (DMEM) supplemented with 10% fetal bovine serum (FBS) and 1% antibiotic-antimycotic (Gibco, Waltham, MA, USA) and maintained at 37 °C in a humidified incubator with 5% CO_2_. Cells were subcultured at approximately 80% confluency and routinely tested for mycoplasma contamination using PlasmoTest (InvivoGen, San Diego, CA, USA).

### 4.3. Cell Proliferation Assay

Cell proliferation was evaluated using the Cell Counting Kit-8 (CCK-8, Dojindo, Japan), following the manufacturer’s protocol. Briefly, A375 EVCtrl and GCNT2^OE^ cells were seeded in 96-well plates (2000 cells/well), allowed to adhere overnight, and treated with either 0.1 μg/mL rhGal-3 or vehicle control. Proliferation was assessed at 0, 24, 48, and 72 h by adding 10 μL of CCK-8 reagent to each well and incubating at 37 °C for 2 h. Absorbance at 450 nm was measured using a Cytation 5 reader (BioTek, Winooski, VT, USA). All conditions were tested in triplicate, and experiments were performed independently at least three times.

### 4.4. Transwell Migration Assay

Cell migration was assessed using an 8.0 μm pore Transwell system (Corning, Corning, NY, USA) as previously reported [37]. Briefly, cells (5  ×  10^5^) in 200 μL serum-free medium, with or without 0.1 μg/mL rhGal-3, were seeded into the upper chamber. The lower chamber contained medium with 30% FBS as a chemoattractant. After 16 h, non-migrated cells were removed with a cotton swab, and migrated cells on the underside of the membrane were fixed with 4% paraformaldehyde and stained with 0.5% crystal violet. Migrated cells were counted in five random fields per well at 40× magnification using an EVOS FL Imaging System (Life Technologies, Carlsbad, CA, USA). Data represent mean ± SEM from three independent experiments.

### 4.5. RT-qPCR Analysis

A375 EVCtrl and GCNT2^OE^ cells were cultured in complete medium, with or without 0.5 μg/mL rhGal-3 for 48 h. Total RNA was isolated using the RNeasy Plus Mini Kit (Qiagen, Hilden, Germany) and reverse-transcribed into cDNA with the SuperScript VILO cDNA Synthesis Kit (Invitrogen, Carlsbad, CA, USA). Quantitative PCR was performed using TaqMan Fast Advanced Master Mix (Applied Biosystems, Waltham, MA, USA) and TaqMan probes for BCL2, BCL2L1, BAX, BAD, and GAPDH as the internal control. All reactions followed standard procedures and manufacturer instructions.

### 4.6. Immunoblotting

Immunoblot analysis was conducted as previously outlined [37]. Briefly, A375 EVCtrl and GCNT2^OE^ cells were serum-starved overnight, and then treated with varying concentrations of rhGal-3 (0, 0.02, 0.1, 0.5, and 1 μg/mL) for 30 min in serum-free medium. Cells were lysed in RIPA buffer containing protease and phosphatase inhibitors. Protein concentration was determined using the BCA assay (Thermo Scientific, Waltham, MA, USA), and equal amounts were prepared in Laemmli buffer, boiled, and separated by 4–12% SDS-PAGE (BioRad, Hercules, CA, USA). Proteins were transferred to PVDF membranes, blocked, and incubated overnight at 4 °C with primary antibodies against *p*-ERK1/2, total ERK1/2, and β-actin, followed by IRDye secondary antibodies. Detection and analysis were performed using a LI-COR imaging system.

### 4.7. Gal-3 Affinity Chromatography

A375 EVCtrl and GCNT2^OE^ cells were detached with Accutase (Fisher), washed with PBS, and lysed in IP lysis buffer (Pierce). Protein concentrations were determined using a Pierce BCA protein assay kit (Thermo Scientific). At least 50 μg of lysate was incubated with 10 μg/mL rhGal-3 (Peprotech) for 30 min on ice. The reaction mixture was then incubated overnight with 2 μg of anti-Gal-3 antibody (Biolegend, San Diego, CA, USA) at 4 °C with mixing. Gal-3 binding partners were immunoprecipitated using Pierce Protein G Magnetic Beads and eluted using Pierce IgG Elution Buffer per the manufacturer’s protocol. Gal-3 binding partners were collected and analyzed via immunoblotting as mentioned above using anti-β1 integrin antibody (Cell Signaling). Lactose (100 mM) was used as a control to compete with Gal-3 binding and confirm the carbohydrate-dependence of the interaction.

### 4.8. Flow Cytometry

Cells were harvested, washed, and incubated with 10 μg/mL recombinant Gal-3 (Peprotech) for 45 min on ice. After washing, the cells were stained with Alexa Fluor 647-conjugated anti-Gal-3 antibody (Biolegend) and Aqua Live/Dead stain for 30 min on ice. Appropriate controls were included. Data were acquired using a FACSCelesta (BD Biosciences, San Jose, CA, USA) and analyzed with the FlowJo software version 10.8.1 (Tree Star, Ashland, OR, USA). The predilection expression site of Gal-3-binding glycans was assessed by analyzing Gal-3 binding to A375 EVCtrl and GCNT2^OE^ melanoma cells following 48 h treatment with the N-glycosylation inhibitor Kifunensine. Median fluorescence intensity (MFI) for Gal-3 binding was determined for both untreated (control) and kifunensine-treated cells using FlowJo software. The fold-change in Gal-3 binding after treatment was calculated by normalizing the MFI of treated samples to the untreated control, which was set to 100%. This calculation is expressed asFold change = (MFI_Kifunensine_/MFI_Control_) × 100

### 4.9. Statistical Analysis

Statistical analyses were performed using Prism 10.0 software (GraphPad). For comparisons between two groups, Student’s unpaired two-tailed *t*-test was used for normally distributed data, while the Mann–Whitney test was applied for non-normally distributed data; normality was assessed using the Shapiro–Wilk test. For data involving multiple groups and/or time points, two-way ANOVA followed by Tukey’s multiple comparisons test was employed. All experiments were performed with a minimum of three independent replicates. Data are presented as means  ±  SEM unless otherwise noted. A *p*-value < 0.05 was considered statistically significant.

## Figures and Tables

**Figure 1 ijms-26-04882-f001:**
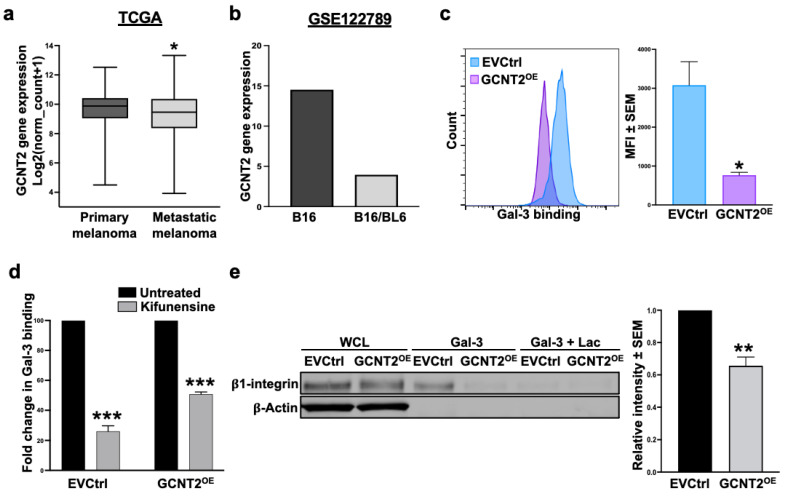
GCNT2/I-branching impedes Gal-3 binding to the melanoma cell surface. GCNT2 expression in metastatic melanomas (*n* = 368) versus primary melanomas (*n* = 103) retrieved from the TCGA SKCM database (**a**) and the GEO RNA-seq dataset GSE122789 (*n* = 1 for each cell line) (**b**). Flow cytometry analysis of rhGal-3 binding to A375 EVCtrl and GCNT2^OE^ cells (**c**). Flow cytometry analysis of rhGal-3 binding to EVCtrl and GCNT2^OE^ cells following 48 h treatment with Kifunensine (**d**). Gal-3 affinity chromatography followed by β1 integrin immunoblotting in EVCtrl and GCNT2^OE^ cells (**e**). (*** *p* < 0.001, ** *p* < 0.01, * *p* < 0.05).

**Figure 2 ijms-26-04882-f002:**
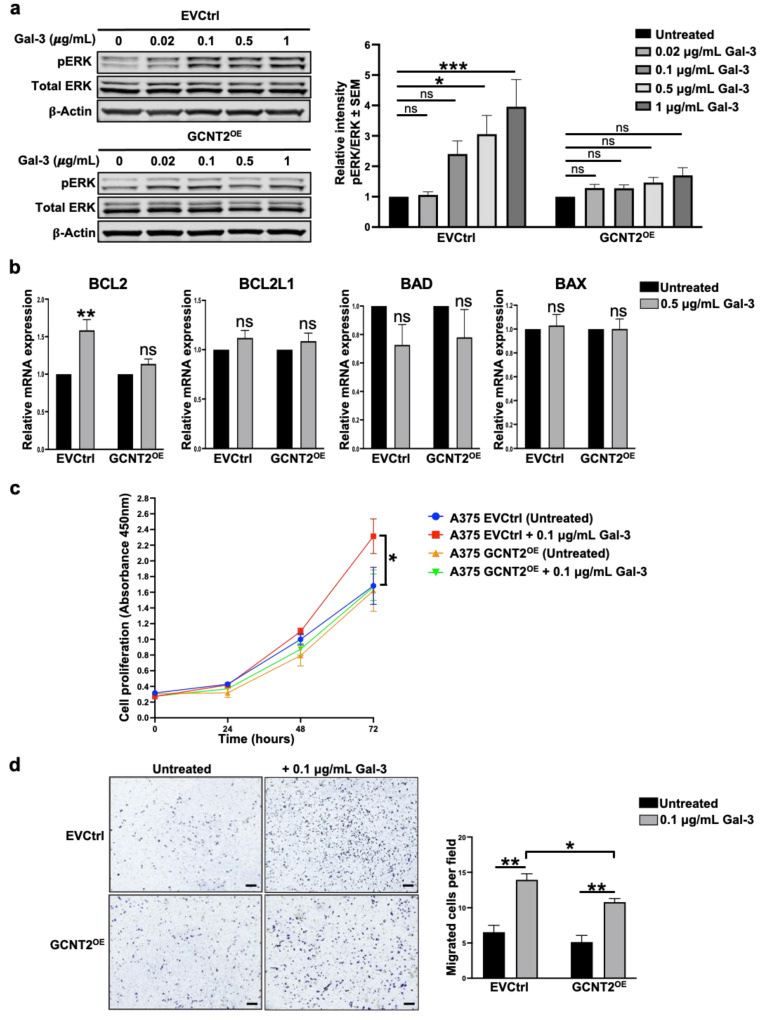
GCNT2/I-branching modulates Gal-3-dependent oncogenic signaling and cell behavior in melanoma. Immunoblot analysis of MAPK pathway activation in A375 EVCtrl and GCNT2^OE^ cells at 0, 0.02, 0.1, 0.5, and 1 μg/mL rhGal-3 (**a**). RT-qPCR analysis of pro-survival (BCL2 and BCL2L1) and pro-apoptotic (BAD and BAX) gene expression in A375 EVCtrl and GCNT2^OE^ cells at 0.5 μg/mL rhGal-3 (**b**). In vitro proliferation ability of A375 EVCtrl and GCNT2^OE^ cells at 0.1 μg/mL rhGal-3 (**c**). Transwell migration ability of A375 EVCtrl and GCNT2^OE^ cells at 0.1 μg/mL rhGal-3 (scale bar = 250 µm) (**d**). (*** *p* < 0.001, ** *p* < 0.01, * *p* < 0.05; ns, not significant).

**Figure 3 ijms-26-04882-f003:**
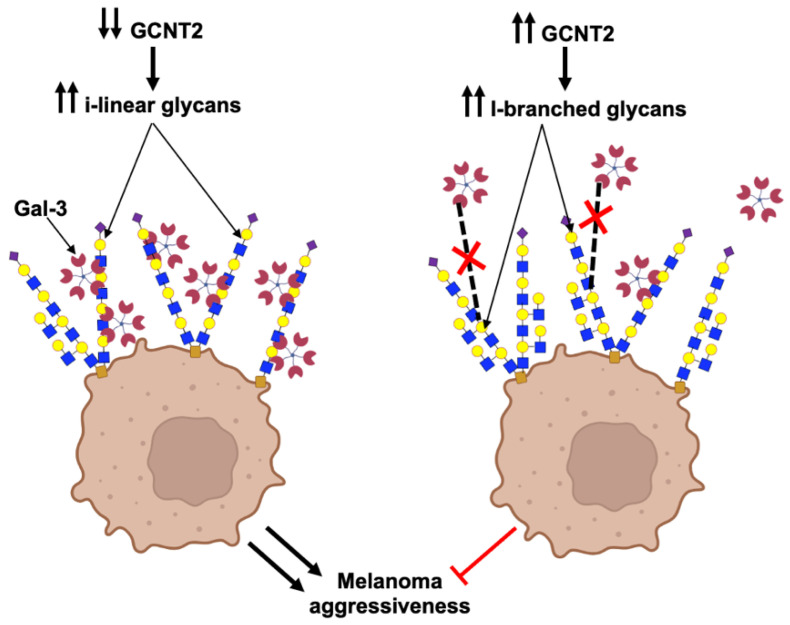
GCNT2/I-branching inhibits Gal-3-binding to melanoma cells and suppresses related pro-oncogenic activities. This schematic depicts the effects of I-branches synthesized by GCNT2 on Gal-3 binding and subsequent cell behavior in melanoma. In cells with low GCNT2 expression (**left** panel), the predominance of i-linear poly-LacNAc chains on cell surface glycoconjugates facilitates Gal-3 binding, leading to enhanced melanoma aggressive behavior. Conversely, in cells with high GCNT2 expression (**right** panel), the formation of I-branched poly-LacNAcs on cell surface receptors impairs Gal-3 binding, resulting in attenuated melanoma aggressiveness. (Red X indicates inhibited Gal-3 binding) (Created using the BioRender illustrating tool (Biorender.com) (accessed on 15 May 2025)).

## Data Availability

The data analyzed in this study were obtained from The Cancer Genome Atlas (TCGA) database, which is publicly available through the Genomic Data Commons (GDC) Data Portal (https://portal.gdc.cancer.gov/, accessed on 15 May 2025) and the Gene Expression Omnibus (GEO) database at GSE122789.

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
