# Peer review of "Melanoma Glycome Regulates the Pro-Oncogenic Properties of Extracellular Galectin-3"

_ijms, 2025, doi:10.3390/ijms26104882_

Round 1

Reviewer 1 Report

Comments and Suggestions for Authors

Revision of manuscript ID ijms-3553842 titled “Melanoma glycome regulates the pro-oncogenic properties of 2 extracellular galectin-3” by Mohammed et al, to the journal International Journal of Molecular Sciences.

In this study, the authors focused on the role of GCNT2 overexpression in reducing the effects promoted by the exogenous addition of Gal-3 in A375 melanoma cell line model. In addition, they used in-silico data to demonstrate that GCNT2 decreased in metastatic melanoma.

The study is well-written, and the data is organized and well-presented. Having a graphical abstract in Figure 2 helps to understand the rationale of the study. However, I have several concerns about the reproducibility of the current results in other metastatic melanoma cell models. In the current study, the authors only used A375 melanoma cell line and ran into the risk that the results are representative of just one cell line model. I also have some statistical concerns that need to be addressed. Overall, the study should be improved before being considered.

Major comments

  1. The authors should reproduce the results obtained in A375 in another cell line to strengthen the conclusions and the applicability of the biological findings to more than one in-vitro model.
  2. The authors need to determine the levels of cell proliferation in EVCtrl and GCNT2-OE cell lines in the presence of Gal-3. This will demonstrate whether the ERK activation results in higher rates of proliferation.
  3. The increase in the BCL2mRNA levels is modest, but significant. However, demonstrating that there is an increase in the BCL2 protein levels will improve the results and conclusions.
  4. The images of the migration assay should be improved as the current images are of low quality. Also, include the scale bars in all the images.
  5. In figure 1f, there are more than two groups included in the analysis and therefore, t-test or Mann-Whitney are not the proper test. Please review the statistical analysis.
  6. Although, I understand that GCNT2 overexpression may affect the I-branching in many receptors, the results of the present study will significantly strengthen from performing siRNA to downregulate β1-integrin and evaluate the downstream effects of Gal-3 in activating ERK.

Minor comments

  1. Please add the statistical significance in Figure 1b. Also indicate how many samples were included in the analysis.
  2. All the Figures will benefit from indicating the number of experimental replicates or the number of patients included in the analysis. That can be indicated in the legends and/or in the Figures.
  3. For Figure 1d, I could not find the description of the analysis in M&M. Also, I am not sure how the authors evaluated the fold-change in Gal-3 binding. Please add and describe properly.
  4. The supplementary figures showing the uncropped WB images would greatly benefit from adding the MW markers.

Author Response

Major comments

Comment 1: The authors should reproduce the results obtained in A375 in another cell line to strengthen the conclusions and the applicability of the biological findings to more than one in-vitro model.

Response 1: We appreciate the reviewer’s insightful suggestion and agree that validating our findings in additional cell lines would further strengthen the conclusions. However, as this manuscript is a short communication, our primary goal is to provide meaningful data to highlight novel observations and generate interest for further investigation. We plan to expand upon these findings in future studies, and we believe that the current data are valuable for the scope of this communication.

Comment 2: The authors need to determine the levels of cell proliferation in EVCtrl and GCNT2-OE cell lines in the presence of Gal-3. This will demonstrate whether the ERK activation results in higher rates of proliferation.

Response 2: We thank the reviewer for their comment and recommendation. In response, we performed an in vitro proliferation assay with and without Gal-3 treatment, and have updated the manuscript as follows:

  • We have added the experimental approach to the Materials and Methods (Pages 6-7) as “Cell proliferation assay. Cell proliferation was evaluated using the Cell Counting Kit-8 (CCK-8, Dojindo, Japan), following the manufacturer’s protocol. Briefly, A375 EVCtrl and GCNT2OE cells were seeded in 96-well plates (2,000 cells/well), allowed to adhere overnight, and treated with either 0.1 μg/mL rhGal-3 or vehicle control. Proliferation was assessed at 0, 24, 48, and 72 hours by adding 10 μL of CCK-8 reagent to each well and incubating at 37°C for 2 hours. Absorbance at 450 nm was measured using a Cytation 5 reader (BioTek). All conditions were tested in triplicate, and experiments were performed independently at least three times.”
  • We have incorporated the proliferation data into the revised Figure 2.c (Page 13) and stated in the Results section (Page 14) that “To further investigate the functional effects of Gal-3 treatment on melanoma cells and how they are modulated by I-branching, we evaluated the proliferative abilities of Gal-3-treated cells. Gal-3 treatment significantly increased cell proliferation over time (p<0.001), with a significant interaction between cell line, treatment, and time (p<0.001), indicating that the proliferative response to Gal-3 differed between groups. At 72 hours, Gal-3-treated A375 EVCtrl cells showed a significantly higher proliferation compared to untreated A375 EVCtrl cells (p<0.05) (Figure 2c). In contrast, Gal-3 treatment did not significantly enhance proliferation in A375 GCNT2OE cells at any time point (Figure 2c). These findings suggest that Gal-3 preferentially enhances the proliferation of melanoma cells exhibiting reduced levels of surface GCNT2/I-branched glycans.”

Comment 3: The increase in the BCL2mRNA levels is modest, but significant. However, demonstrating that there is an increase in the BCL2 protein levels will improve the results and conclusions.

Response 3: We appreciate the reviewer's suggestion to include BCL2 protein level analysis. While protein data can offer additional layers of confirmation, it is well established that qPCR-based quantification of BCL2 mRNA is a widely accepted and reliable method for assessing BCL2 expression [1], providing a valuable insight into the cellular response being studied. Considering that our primary aim is to demonstrate the transcriptional changes triggered by Gal-3 and how this is affected by the abundance of surface I-branches, we believe that the significant increase in BCL2 mRNA is both meaningful and sufficient for the scope of this short communication.

Comment 4: The images of the migration assay should be improved as the current images are of low quality. Also, include the scale bars in all the images.

Response 4: We thank the reviewer for their comment. We have improved the quality of the images and included a scale bar in all of them (Page 13_Figure 2.d).

Comment 5: In figure 1f, there are more than two groups included in the analysis and therefore, t-test or Mann-Whitney are not the proper test. Please review the statistical analysis.

Response 5: We thank the reviewer for their comment. We have revised the analysis and now report the use of a Two-Way ANOVA as follows:

  • Materials and Methods (Page 9): We now stated that “For data involving multiple groups and/or time points, two-way ANOVA followed by Tukey's multiple comparisons test was employed.”
  • Results (Page 12) and Figure 2.a (Page 13): We updated Figure 2.a and stated that “Data analysis using a two-way ANOVA revealed a significant interaction between cell line and Gal-3 treatment on ERK phosphorylation (pERK) (p<0.05). Post-hoc tests showed that in A375 EVCtrl cells, Gal-3 significantly increased ERK phosphorylation at concentrations of 0.5 μg/mL (p<0.05) and 1 μg/mL (p<0.001) (Figure 2a). Conversely, in A375 GCNT2OE cells, Gal-3 treatment did not significantly alter ERK phosphorylation at any tested concentration (p>0.05) (Figure 2a). These results indicate that Gal-3 more effectively stimulates ERK phosphorylation in A375 EVCtrl cells compared to A375 GCNT2OE cells, suggesting that the presence of GCNT2/I-branching modulates the intracellular response of the MAPK signaling axis to the exogenous binding of Gal-3 to melanoma cells.

Comment 6: Although, I understand that GCNT2 overexpression may affect the I-branching in many receptors, the results of the present study will significantly strengthen from performing siRNA to downregulate β1-integrin and evaluate the downstream effects of Gal-3 in activating ERK.

Response 6: We appreciate the reviewer's insightful suggestion. While we agree that this would be a valuable experiment to further elucidate the mechanisms involved, we consider this to be an important future direction for our research. Given the scope of the current manuscript as a short communication, these additional experiments are beyond what we aimed to present here, but we will consider them for subsequent, more extensive studies.

Minor comments

Comment 1: Please add the statistical significance in Figure 1b. Also indicate how many samples were included in the analysis.

Response 1: We thank the reviewer for their comment. The data presented in Figure 1b for Gal-3 expression in the two murine cell lines (B16 and B16/BL6) were derived from the GSE122789 dataset, which contains a single sample for each cell line. Consequently, standard statistical tests for significance are not applicable for this comparison. We have updated the figure legend (Page 10) to explicitly state the number of samples included in the analysis as “n=1 for each cell line”.

Comment 2: All the Figures will benefit from indicating the number of experimental replicates or the number of patients included in the analysis. That can be indicated in the legends and/or in the Figures.

Response 2: We thank the reviewer for this valuable suggestion. In response, we have updated figure legends to indicate the number of patients and cell lines used in each analysis. We have also stated that “All experiments were performed with a minimum of three independent replicates” in the Methods section under “Statistical analysis” (Page 9).

Comment 3: For Figure 1d, I could not find the description of the analysis in M&M. Also, I am not sure how the authors evaluated the fold-change in Gal-3 binding. Please add and describe properly.

Response 3: We thank the reviewer for pointing this out. The description of the analysis for Figure 1d has been added to the Methods section (Page 8-9) as follows:

Flow cytometry. Cells were harvested, washed, and incubated with 10 μg/mL recombinant Gal-3  (Peprotech) for 45 minutes on ice. After washing, cells were stained with Alexa Fluor 647-conjugated anti-Gal-3 antibody (Biolegend) and Aqua Live/Dead stain for 30 minutes on ice. Appropriate controls were included. Data were acquired using a FACSCelesta (BD Biosciences, San Jose, CA, USA) and analyzed with the FlowJo software (Tree Star, Ashland, OR, USA). The predilection expression site of Gal-3-binding glycans was assessed by analyzing Gal-3 binding to A375 EVCtrl and GCNT2OE melanoma cells following 48h treatment with the N-glycosylation inhibitor Kifunensine. Median fluorescence intensity (MFI) for Gal-3 binding was determined for both untreated (control) and kifunensine-treated cells using FlowJo software. The fold-change in Gal-3 binding after treatment was calculated by normalizing the MFI of treated samples to the untreated control, which was set to 100%. This calculation is expressed as:

Fold change = (MFIKifunensine/ MFIControl) ×100

Comment 4: The supplementary figures showing the uncropped WB images would greatly benefit from adding the MW markers.

Response 4: We thank the reviewer for this suggestion. The supplementary figures have been updated to include the molecular weight (MW) markers on all WB images.

Reference:

  1. Placzek, W., et al., A survey of the anti-apoptotic Bcl-2 subfamily expression in cancer types provides a platform to predict the efficacy of Bcl-2 antagonists in cancer therapy. Cell death & disease, 2010. 1(5): p. e40-e40.

Reviewer 2 Report

Comments and Suggestions for Authors

The manuscript established the inverse correlation of GCNT2 expression and the metastatic capacity of melanoma and showcased its I-branching hinders Gal-3 binding to its glycoprotein receptor β1 integrin and attenuates the pro-migratory effects of extracellular Gal-3 on melanoma cells. This work is well organized and supported by quality data. 2 Minor comments are listed below:

(1) Page 4, Figure 1c, the color scheme can be homogenized;

(2) Page 4, Figure 1g, 'BAX' mislabelled with 'BAD'.

Author Response

Comment 1: Page 4, Figure 1c, the color scheme can be homogenized;

Response 1: We thank the reviewer for their comment. We have homogenized the colors of Figure 1c in the revised manuscript (Page 10).

Comment 2: Page 4, Figure 1g, 'BAX' mislabelled with 'BAD'.

Response 2: We thank the reviewer for their comment. We have corrected the mislabeled graph in the revised manuscript, which is now Figure 2.b (Page 13).

Reviewer 3 Report

Comments and Suggestions for Authors

The data presented in the manuscript by Mohammed et al. is incremental research on the role of GCNT2-mediated glycosylation in melanoma progression. Effects of Galectin-3 binding to linear or branched glycans and the consequences thereof on melanoma cells in vitro are presented concisely and in a well-structured manner and technically manuscript has no flaws. It is the Editor's discretion to decide whether the data is substantial enough for publication in the IJMS.

Minor: It may be useful for the ease of reading to split the Figure 1 into two figures and separate the results from section 2.3 onwards into a separate figure.

Author Response

Comment 1: Minor: It may be useful for the ease of reading to split the Figure 1 into two figures and separate the results from section 2.3 onwards into a separate figure.

Response 1: We thank the reviewer for their comment. We have split Figure 1 into two figures: Figure 1 (Page 10) and Figure 2 (Page 13).

Round 2

Reviewer 1 Report

Comments and Suggestions for Authors

The authors addressed mostly all of my comments. I agree with their comments that some of the experiments requested, are beyond of the scope of this communication to IJMS.